# Echocardiography Imaging of the Right Ventricle: Focus on Three-Dimensional Echocardiography

**DOI:** 10.3390/diagnostics13152470

**Published:** 2023-07-25

**Authors:** Andrea Ágnes Molnár, Attila Sánta, Béla Merkely

**Affiliations:** Heart and Vascular Center, Semmelweis University, 1085 Budapest, Hungary; santa.attila99@gmail.com (A.S.); merkely.bela@gmail.com (B.M.)

**Keywords:** three-dimensional, right ventricle, tricuspid valve, ejection fraction, predictive value

## Abstract

Right ventricular function strongly predicts cardiac death and adverse cardiac events in patients with cardiac diseases. However, the accurate right ventricular assessment by two-dimensional echocardiography is limited due to its complex anatomy, shape, and load dependence. Advances in cardiac imaging and three-dimensional echocardiography provided more reliable information on right ventricular volumes and function without geometrical assumptions. Furthermore, the pathophysiology of right ventricular dysfunction and tricuspid regurgitation is frequently connected. Three-dimensional echocardiography allows a more in-depth structural and functional evaluation of the tricuspid valve. Understanding the anatomy and pathophysiology of the right side of the heart may help in diagnosing and managing the disease by using reliable imaging tools. The present review describes the challenging echocardiographic assessment of the right ventricle and tricuspid valve apparatus in clinical practice with a focus on three-dimensional echocardiography.

## 1. Introduction

Right ventricular (RV) dysfunction is associated with the worse outcome of several cardiac diseases including heart failure, cardiomyopathies, right ventricular myocardial infarction, congenital heart diseases, pulmonary arterial hypertension, and valvular heart diseases [1,2,3,4,5,6,7,8,9,10]. The assessment of right ventricular function by two-dimensional (2D) echocardiography remains challenging because of its complex anatomy, shape, and significant load dependence [1]. Advances in three-dimensional (3D) echocardiography imaging, including machine learning algorithms, provide a more reliable and feasible examination of the right ventricle without geometrical assumptions [1,11]. Furthermore, the pathophysiology of RV function is often related to tricuspid regurgitation (TR) [12,13]. A mild degree of TR is highly prevalent and is usually considered a benign incidental finding on routine echocardiography [12]. However, moderate to severe TR is related to worsening survival regardless of pulmonary hypertension (PH) and left ventricular systolic function, which has raised interest in the diagnostics and management of TR [14,15,16]. Enface 3D views of the tricuspid valve can supplement 2D echocardiography measurements and provide further valuable data regarding leaflets, annular size, etiology, and severity of valve regurgitation [17]. 

Understanding the anatomy and pathophysiology of the right side of the heart by using competent imaging tools may help in diagnosing and managing the disease [18,19]. Knowledge of the complex anatomy of the right ventricle allows us to interpret the conventionally used 2D echocardiography parameters more properly and to be convinced of the usefulness of 3D echocardiography in estimating RV size and function. Most of the 2D echocardiography RV functional parameters used in daily clinical practice are one-dimensional parameters, which cannot accurately estimate global RV function. Furthermore, knowledge of RV pathophysiological states with the application of appropriate imaging tools could result in a more accurate diagnosis and optimal patient management. Recently, onsite 3D echocardiography RV analysis became more available in echocardiography laboratories. It is considered less time-consuming, even in routine clinical practice, due to the novel software using artificial intelligence, which has revolutionized data processing and interpretation. However, high-quality image acquisition is still a cornerstone of 3D RV analysis and poor 2D echocardiography image quality cannot be replaced by 3D echocardiography examination.

This review aimed to provide a short overview of the right ventricular and tricuspid valve anatomy, pathophysiology, and 2D/3D echocardiography assessment to demonstrate in detail the added value of 3D echocardiography using the latest published data besides the state-of-the-art literature.

## 2. Anatomy and Physiology of the Right Ventricle and Tricuspid Valve

The anatomical structure of the right ventricle is complex [1,20,21]. The three components of the RV have different embryological origins and electrophysiological properties [22]. The inlet component constitutes the tricuspid valve apparatus, and the trabecular component involves the apex with the three intracavitary muscle bands (crista supraventricularis, septomarginal trabeculation, and moderator band), and the outlet component includes the subpulmonary infundibulum [1,20,21]. (Figure 1) The right ventricle is remarkably thinner than the left ventricle (approximately 5 mm in adults) [20,23]. It contains 30% more collagen, and the cardiomyocytes are 15% smaller compared to the left ventricle [20,23,24]. It is adopted to the high-compliance, low-resistance pulmonary circuit [20]. The RV myocardium consists of two layers [20]. The superficial and thinner circumferential layer is set parallel to the atrioventricular groove extending to the left ventricle; meanwhile, the predominant subendocardial layer with longitudinally arranged myocytes passes through the apex toward the tricuspid annulus and outflow [20]. The biventricular functional interdependence is attributable not only to the septum, but to the shared epicardial circumferential myocytes [20,25]. The circumferential fibers contribute mainly to the bellow-like contraction of the right ventricle [26]. Overall, the contraction of the right ventricle is a peristalsis-like motion as the contraction starts earlier within the inlet and trabeculated myocardium than the outlet myocardium, which serves as a pressure buffer while transmitting the blood flow to the pulmonary artery [20]. 

The right ventricle and tricuspid valve are tightly related because the tricuspid annulus is functionally a part of the right ventricle [13]. The anatomy of the tricuspid valve is largely variable [17]. Hahn and coworkers [17] found that the most common anatomic variant is the classic three-leaflet morphology tricuspid valve, which occurs in 28% to 58% of cases. The second most common anatomic subtype is the quadricuspid morphology tricuspid valve, which was observed in 39% of the cases [17]. Typically, the tricuspid valve leaflets are of unequal size. Usually, the anterior and septal leaflet is the largest, and the posterior leaflet is smaller [17]. The septal leaflet is displaced approximately 10 mm apically to the septal insertion of the anterior mitral leaflet. The normal tricuspid annulus is a dynamic, non-planar, oval, saddle-shaped fibrous component of the tricuspid valve apparatus, which is considered a component of the right ventricle [13]. The shape and size of the tricuspid annulus highly depend on loading conditions, as the fibrous annulus is incomplete at the RV-free wall region, allowing potential dilatation at this location [17,27]. The anterior papillary muscle supports the chordae of the anterior and posterior leaflet, while the posterior papillary muscle supports the chordae of the posterior and septal leaflet. The septal papillary muscle is variable, and it may be absent in up to 20% [28]. The length of the chordae is fixed; consequently, enlargement of the right ventricle may affect tricuspid leaflet coaptation [13]. 

## 3. Pathophysiological Aspects of the Right Ventricle and Tricuspid Valve

Right ventricular dysfunction is the consequence of RV pressure overload, volume overload, or impaired myocardial contractility [20]. The RV wall is thinner and less muscular compared to the left ventricular wall. Consequently, the distensibility and capacity of the right ventricle are higher. This results in a greater RV adaptation to volume overload rather than pressure overload [20]. 

### 3.1. Acute Right Ventricular Pressure and Volume Overload 

Acute right heart failure is mainly a result of acute pressure overload (e.g., acute massive pulmonary embolism) or acute impairment of myocardial contractility (e.g., acute right ventricular myocardial infarction) [23,29,30]. Acute volume overload (e.g., acute TR in complicated myocardial biopsy) rarely leads to acute heart failure. The pressure overload of acute massive pulmonary embolism may rapidly decrease RV stroke volume leading to hemodynamic collapse [23]. Isolated RV myocardial infarction is rare. In acute RV myocardial infarction, the right ventricle becomes stiff, resulting in increased diastolic pressure and reduction in blood flow from right atria; therefore, the right ventricle delivers less blood to the left side of the heart, leading to low cardiac output and cardiogenic shock, even in the presence of normal left ventricular function [29,30]. Furthermore, the dilated right ventricle shifts the interventricular septum towards the left side, further impairing left ventricular filling [23,29,30]. 

### 3.2. Chronic Right Ventricular Pressure Overload 

Pulmonary arterial pressure (PAP) and pulmonary vascular resistance (PVR) increase slowly in chronic pressure overload, and consequently the stimulated myocytes lead to adaptive hypertrophy to preserve cardiac output [31]. Pulmonary hypertension is concluded when the mean pulmonary arterial pressure (mPAP) is ≥20 mmHg, as confirmed by right-sided heart catheterization [32]. The pressure overload-induced concentric RV hypertrophy is characterized by preserved volumes and function with increased right ventricular mass in the compensated phase [20,31,33]. In the decompensated stage, when contractility can no longer compensate pressure overload, eccentric RV hypertrophy takes place, with progressive dilatation and increase in filling pressures [20,31,33]. The prolonged RV contraction leads to interventricular dyssynchrony and leftward septal shift, which results in left ventricular underfilling [20,31,34]. Pulmonary hypertension is classified into five clinical subgroups: pulmonary arterial hypertension (PAH, group 1), pulmonary hypertension due to left-sided heart disease (PH-LHD, group 2), pulmonary hypertension associated with chronic lung disease and/or hypoxia (PH-CLD, group 3), pulmonary hypertension associated with chronic pulmonary artery obstruction (CTEPH, group 4), and pulmonary hypertension due to unclear and/or multifactorial mechanisms (group 5) [35]. The latter group encompasses hematologic disorders, systemic and metabolic disorders, fibrosing mediastinitis or chronic renal failure disease, and complex congenital heart diseases [35]. Precapillary PH with less than 15 mmHg of pulmonary artery wedge pressure (PAWP) and more than 3 Wood units (WU) of pulmonary vascular resistance (PVR) is characteristic for PAH and PH-CLD [32,35]. Postcapillary PH with more than 15 mmHg PAWP and less than 3 WU of PVR is characteristic for PH-LHD; however, combined postcapillary and precapillary PH (PAWP > 15 mm Hg and PVR ≥ 3 WU) can occur. The one-year mortality rate in PH population ranges from 10% to 32%, with the worst mortality rate in the PH associated with the left heart disease group [36,37]. Nonetheless, the five-year mortality rate can even reach 69% in the PH due to chronic lung disease population [38].

### 3.3. Chronic Right Ventricular Volume Overload 

Chronic RV volume overload develops mainly in TR, pulmonary regurgitation, and left-to right congenital shunts such as atrial septal defects (ASD) [39,40,41]. Tricuspid regurgitation is a common finding on echocardiography examination. According to the tricuspid leaflet involvement, TR is classified into primary (organic), secondary (functional), and cardiac implantable electronic device (CIED)-related regurgitation [17,42]. Primary TR is characterized by the pathology of the tricuspid valve and/or subvalvular apparatus due to congenital or acquired etiology. Epstein’s anomaly is one of the most common congenital anomalies affecting the tricuspid valve and is characterized by displacement of the leaflets towards the apex [43]. Acquired causes comprise endocarditis, prolapse, connective tissue disorder, systemic diseases, radiation, rheumatic disease, tumors, and drug-induced leaflet damage [17,44]. Functional TR corresponds to non-leaflet pathology which represents the predominant mechanism of TR. It develops as a consequence of tricuspid annulus dilatation due to right atrial enlargement (atrial functional TR) or right ventricular remodeling (ventricular functional TR) [17]. Atrial functional TR is defined by permanent atrial fibrillation in the absence of tricuspid leaflet pathology, RV dysfunction, left-sided heart disease, and PH [45]. Ventricular functional TR develops in RV dysfunction with or without PH [17]. The most common cause of ventricular functional TR is left-sided heart disease, usually chronic mitral regurgitation followed by aortic stenosis [46,47]. CIED-related TR represents a distinct and unique etiology compared to the other valves [17,48]. There are two pathways leading to CIED-related TR. Primary CIED-related TR occurs when the pacemaker lead interferes with the tricuspid valve apparatus. The pacemaker lead may perforate or lacerate the leaflets and damage the papillary muscles and chordae tendineae [48]. Nevertheless, secondary CIED-related TR occurs when the pacing itself leads to heart failure and RV enlargement. Notably, primary and secondary CIED-related TR can overlap [17,48]. Pulmonary regurgitation is less common compared to TR; however, it frequently occurs after repaired TOF as a consequence of RV outflow tract remodeling [39,49]. Overall, volume overload is better tolerated compared to pressure overload due to the biomechanical properties of the thinner right ventricular wall with higher distensibility. Eccentric hypertrophy and predominant diastolic leftward interventricular septal shift are the most common features of right ventricular volume overload. Right ventricular contractility remains preserved; however, contractile reserve can be diminished [20]. In significantly elevated PVR and/or RV dysfunction, correction of TR must be considered with caution [35,50]. 

### 3.4. Intrinsic Right Ventricular Myocardial Disease 

The etiology of intrinsic RV myocardial disease involves myocardial ischemia, myocarditis, cardiotoxicity, arrhythmogenic RV cardiomyopathy (ARVC), hypertrophic cardiomyopathy, amyloidosis, and sarcoidosis [20]. Isolated RV infarction is rare; however, RV involvement in acute left ventricular myocardial infarction occurs in up to 50% of cases [51,52]. The prevalence of chronic scars after the acute phase of right ventricular infarction is relatively low, suggesting that post-infarction RV dysfunction is multifactorial in etiology [20,53]. Patients with inferior myocardial infarction and RV involvement have a higher mortality risk compared to patients without RV involvement [52,54]. In ARVC, the fibrofatty replacement of cardiomyocytes is present mainly at the infundibulum, subtricuspid region, and apex [20,55]. Furthermore, cardiotoxicity-induced right ventricular dysfunction might represent a future diagnostic and clinical issue, due to the increasing number of cancer survivors [56]. The RV wall is thinner compared to the left ventricular wall, without potential for restitution, and is thus more susceptible to cardiotoxicity. Nonetheless, most of the publications focus mainly on left ventricular dysfunction, neglecting the potential impact of right ventricular function [56].

## 4. Two-Dimensional Echocardiography of the Right Ventricle and Tricuspid Valve

The routine imaging method of the right ventricle in daily clinical practice is 2D echocardiography. However, due to the complex geometry of RV, many of the 2D parameters are inaccurate, with inherent limitations [26]. Consequently, the prognostic value of 2D echocardiography-derived RV dimensions and function was less evaluated compared to the parameters of the left ventricle. Measurement of RV basal, mid-cavity diameters, RV outflow tract diameters, and base-to-apex length is recommended [57,58]. The largely used tricuspid annular planar systolic excursion (TAPSE) is a simple and reproducible one-dimensional functional parameter showing the longitudinal contraction of the RV-free wall [26,57,58]. Tricuspid annular plane systolic excursion has prognostic value in PH and heart failure [59,60]. Similarly, tissue Doppler imaging (TDI)-derived RV systolic excursion S’-wave velocity is a highly reproducible one-dimensional functional parameter representing the longitudinal component of RV function [57,58]. However, both TAPSE and TDI-derived S’-wave are angle- and load-dependent parameters and overall inaccurate in RV global systolic function assessment (Figure 2). A value of <17 mm for TAPSE and <9 cm/s for TDI-derived S’ wave is considered diminished [57,58,61] (Figure 2). Right ventricular end-systolic area (ESA) and end-diastolic area (EDA) are used to calculate fractional area change (FAC) and are obtained by tracing the ventricular endocardium in the RV-focused apical four-chamber view. It is considered a more accurate parameter of RV function than TAPSE [58,62]. Reduced RV FAC with a value <35% reflects lower RV systolic function. [26,57,58]. Right ventricular index of myocardial performance (RIMP) is a load-dependent parameter, which estimates both RV systolic and diastolic function [61]. The abnormal value of RIMP is <0.55 measured by the tissue Doppler method, and <0.4 by the pulsed Doppler method [26,57,58]. The contractility of RV is measured by RV dp/dt, which represents the rate of change of pressure, and is calculated using the slope of the tricuspid regurgitation Doppler spectrum between 1 and 2 m/s [61]. RV dp/dt of <400 is considered abnormal; nevertheless, it is a rarely used parameter because of its load dependency [26,57,58]. Right isovolumic myocardial acceleration (IVA) is a less load-dependent parameter of RV performance and is calculated by using the TDI-derived peak isovolumic myocardial velocity devided by the time to reach the peak velocity [61]. However, current guidelines do not recommend the use of IVA in clinical routine due to the low sensitivity and large confidence interval for its normal values [57,58]. Two-dimensional speckle tracking echocardiography parameters of RV are less load- and angle-dependent and estimate RV myocardial function more accurately, although temporal resolution is lower. RV-focused four-chamber view is used to measure RV-free wall strain and RV longitudinal strain (average of the three free wall and three septal segments). The cut-off values for RV-free wall strain and longitudinal strain have been established as −23% and −20%, respectively [61]. (Figure 2) RV strain parameters can detect RV dysfunction earlier than conventional parameters and have a prognostic value in PH, heart failure, myocardial infarction, and TR [63,64,65,66,67,68]. The inter-realtionship between RV function and pulmonary artery systolic pressure is essential; therefore, reporting these inter-relationships in daily routine would be important. However, further validation studies are required [69]. The TAPSE/sPAP ratio represents a non-invasive measure of RV–PA coupling, which might help in the diagnosis of PH [35,70,71]. The mid-systolic ‘notching’pattern of RV outflow tract blood flow may suggest pre-capillary PH [72]. 

The anterior localization of the tricuspid annulus in the mediastinum makes its visualization acceptable by transthoracic echocardiography. The tricuspid annulus consists of fibrous tissue, which is sensitive to preload, afterload, and right ventricular and atrial dilatation [13,27]. The normal diameter of tricuspid annulus in adults is 28 ± 5 mm by 2D-echocardiography measured from apical four-chamber view [73]. Tricuspid annulus dilatation is considered when the mediolateral diameter is >40 mm (>21 mm/m^2^) [73]. The shape and size of tricuspid annulus can change according to loading conditions with clinical impact. However, the two-dimensional evaluation of a non-planar, saddle-shaped 3D structure, such as the tricuspid annulus, can be difficult. The tricuspid annulus is the level where the leaflet coaptation occurs normally with a 5–10 mm coaptation length [74]. The tricuspid annulus dilates in right atrial or right ventricular enlargement. Tricuspid annulus dilatation leads to the tethering of the leaflets with subsequent disappearance of the coaptation length and the appearance of functional tricuspid regurgitation [74,75,76]. The tethering is considered significant when the tethering distance is >8 mm and the tenting area is >1.6 cm^2^, measured by 2D echocardiography [74,75,76]. However, 2D echocardiography presumes that the highest coaptation point is visualized in the apical four-chamber view, which is not always the case [74]. Furthermore, quantification of TR severity by the 2D method can be challenging and requires a multi-parametric approach. Quantitative, semiquantitative, and qualitative assessment of TR severity is recommended by current guidelines [75]. The quantitative evaluation includes effective regurgitant orifice area (EROA) and regurgitant volume (RV) measurement, whereas tricuspid valve inflow, hepatic flow, jet area, annulus dilatation, and vena cava width are semiquantitative parameters [13]. Notably, the proximal isovelocity surface area (PISA) method used for EROA measurement assumes that the regurgitant orifice is flat and circular, and the PISA is hemispherical. This is very unlikely in large, low-velocity tricuspid regurgitant flow [74]. The complex pathological alterations of tricuspid valve geometry usually result in a much more variable shape of regurgitant orifice compared to the mitral regurgitation [74]. Nonetheless, TR is influenced by the loading conditions and respiratory cycle. Qualitative evaluation involves right ventricular size, atrium size and color flow jet examination. [13].

## 5. Three-Dimensional Echocardiography of the Right Ventricle and Tricuspid Valve

The complex geometry of the right ventricle with the inflow and outflow segments in different planes made 2D echocardiography inaccurate by relying on geometrical assumption. Advances in 3D echocardiography imaging made RV and tricuspid valve assessment more detailed and comprehensive, which has been validated against CMR [77,78,79,80,81,82]. Recent software packages have made RV 3D assessment more user-friendly and less time-consuming, even allowing on-site RV evaluation [6]. Furthermore, current guidelines recommend 3D measurement of RV volumes and EF due to its improved accuracy [83]. Full-volume datasets are obtained incorporating inflow, outflow, and apical portion of RV [61,84] (Figure 3). Dedicated software programs are used for post-processing, including RV endocardial surface tracing, which enables accurate volume, function, and shape assessment without geometrical assumptions [61,84]. (Figure 3) However, 3D echocardiography is not a method without limitations. The time and spatial resolution of single capture beat is lower, compared to 2D echocardiography [84]. Consecutive multi-beat acquisition can overcome this limitation by stitching subvolumes together; however, stitching artefacts can occur [84]. The RV is located immediately behind the sternum; therefore, an inadequate image of RV outflow tract and anterior wall is created approximately in 10 to 30% of the cases [1]. A previous meta-analysis showed that right ventricular volumes assessed by 3D echocardiography are slightly underestimated and RV EF slightly overestimated when compared to CMR values [77,78,82,85]. The lower limit of normal 3D RV EF was established as >45% [83]. Muraru and coworkers [6] graded the severity of RV EF according to mortality as mild (45–40%), moderate (40–30%), and severe (<30%) dysfunction. RV EF is independently associated with cardiac and all-cause mortality and major adverse cardiac events (MACE) in patients with different cardiac diseases [86,87]. Normal RV volumes are larger and EF smaller in men than in women, whereas older age (>70 years) was associated with lower RV volumes and higher EF [88,89,90]. Elite athletes represent a special population that is worth mentioning. D’Andrea and coworkers [91] showed that RV end-diastolic volumes were significantly greater in endurance-trained athletes compared to the age- and gender-matched strength-trained athletes and controls. Endurance training leads to the elevation of cardiac output and vagal tone with consequent decrease in afterload, peripheral vascular resistance, and heart rate. These changes result in chronic venous overload, better RV diastolic filling, and increased RV dimensions [91]. These physiologic adaptations to intensive exercise are defined as athlete’s heart [91]. Athlete’s heart must be distinguished from pathologic right heart adaptation to assess sport eligibility [91]. Besides 3D RV EF, 3D speckle tracking echocardiography provides deformation data in three orthogonal planes from one analysis; however, its application is not common in clinical routine [92]. RV area strain is considered a combination of the total vector resultant based on circumferential and longitudinal vectors [5]. 

Measurement of tricuspid annulus diameter by 2D echocardiography from apical four-chamber view represents only the mediolateral size, which does not usually reflect the tricuspid annulus size with its complex geometry accurately [42,93]. In most cases, a change in the anteroposterior diameter can be observed; therefore, the mediolateral diameter does not reflect the extent of tricuspid annular dilatation [93]. In healthy volunteers from the Padua 3D Echo Normal database, Muraru and coworkers [93] confirmed that tricuspid annulus dimensions assessed by 3D echocardiography showed good correlation with cardiac computed tomography measurements. Furthermore, the working group suggested that the dimensions of the tricuspid annulus should be gender-specific and body surface area indexed [93]. The upper limit of normality for tricuspid annulus apical four-chamber view diameter was 42 mm (25 mm/m^2^) for men and 37 mm (23 mm/m^2^) for women by 3D echocardiography [93]. However, the maximal diameter of the annulus irrespective of the orientation was 46 mm (27 mm/m^2^) for men and 43 mm (26 mm/m^2^) for women [93]. The sphericity index of tricuspid annulus decreases during cardiac cycle, leading to the most oval shape at the end-diastole [93]. Quantification of tricuspid regurgitation severity by the 3D planimetry method of vena contracta area measurement might represent a more accurate tool compared to 2D quantification; however, validated cut-off values are still lacking [74]. Earlier publications estimated cut-off vena contracta area values for severe TR in large ranges, from 36 mm^2^ to 75 mm^2^ [94,95,96]. Tricuspid regurgitation usually features different morphologic variance in RV size, tricuspid valve, and annulus anatomy depending on the etiology [47]. The distinction of atrial and ventricular secondary TR using 2D and 3D echocardiography is important due to its prognostic and treatment implications [97,98]. The progression of TR severity is more rapid, and the outcome is worse in atrial functional etiology [17]. Utsunomiya and coworkers [16] found in a large-scale study that in atrial functional TR, the tricuspid annulus dilatation is more prominent and the leaflet tethering is smaller compared to ventricular functional TR. In ventricular functional TR, the RV basal segments were dilated with mild tricuspid annulus dilatation showing conical RV deformation; however, the valve tenting height was excessive [16,46,47].

## 6. Added Value of 3D Echocardiography Compared to 2D Echocardiography in the Assessment of the Right Ventricle

The prognostic value of 3D echocardiography-derived RVEF was proved to be superior to conventional RV echocardiographic parameters for predicting mortality [4,86]. Nagata and coworkers [4] found that 3D RVEF, rather than 3D left ventricular EF, stratified patients from low to high risk for subsequent cardiac events. The RV pump function is composed of the shortening in the longitudinal direction, inward movement of the RV-free wall, and bulging of the interventricular septum into the RV during the left ventricular contraction [99,100]. 

### 6.1. RV Pressure Overload 

It is known that patients with PH have enlarged RV volumes, decreased RV EF, and strain values [92,101,102]. Leary and coworkers [103] found that symptoms, as assessed by the New York Heart Association (NYHA) functional class, were related to 3D echocardiography-derived RVEF and volumes. Nonetheless, 3D RVEF moderately correlated with hemodynamic parameters of right heart catheterization [104]. Liu BY and coworkers [105] showed that 3D RVEF, RV volumes, and RV-free wall strain had independent predictive value in detecting patients stratified in the intermediate-high risk PAH. Similarly, Li Y and coworkers [7] revealed that diminished RVEF increased the risk of clinical progression in CTEPH and showed that patients with 3D RVEF < 30.3% had a poor prognosis. These results coincide with the findings of Ryo and coworkers [106], demonstrating that 3D RV EDVi, RV ESVi and RVEF were associated with poor prognosis in PH. Correspondingly, the stage of pressure-overload-induced RV morphological and functional adaptation is closely related to survival [106]. According to 3D echocardiography-derived RV volumes, patients with PH can be classified into three morphological subsets of RV adaptation and remodeling associated with distinct prognoses, such as RV-adapted, RV adapted–remodeled, and RV adverse–remodeled groups [106]. Routinely used conventional echocardiography parameters, such as PAP, PVR, and TAPSE, were not significantly different between RV adapted–remodeled (compensated) and RV adverse–remodeled (decompensated) groups, but 3D RV ESVi proved to detect the transition to RV decompensation [106]. Furthermore, the working group of Ryo found that 3D RV strain alone was a less sensitive predictor in PH with severe RV dysfunction compared to 3D RV ESVi [106]. In contrast, Moceri and coworkers [5] demonstrated that 3D echocardiography-derived RV strain patterns gradually worsened in PH patients and provided independent prognostic information. In addition, Smith and coworkers [92] showed that the right ventricle became more spherical in PH, and, as a consequence, the direction of subendocardial fibers changed from longitudinal to more circumferential. This leads to reduced composite area strain, which is inversely related to mortality in PH [92]. Interestingly, patients with PH-therapy-induced decrease in PVR have the same worse outcome if the RV function continues to decrease [107]. This suggests that a further RV afterload independent pathway might be important in the progression of PH disease, which could be related to the intrinsic RV myocardial mechanism. 

### 6.2. RV Volume Overload 

Pulmonary hypertension and right ventricular dysfunction often lead to functional tricuspid regurgitation and consequent volume overload [47,108]. Advances in 3D echocardiography imaging allow a more accurate assessment of TR and right ventricle, thus enabling a more in-depth pathophysiological understanding of different TR phenotypes, which may help clinicians make treatment decisions [47] (Figure 4). Notably, 3D echocardiography assessment of TR and right ventricle is still not routinely used in clinical practice, and most of the studies investigating TR outcome used only one- and two-dimensional parameters [15,109]. RV dysfunction and severe TR are also predictors of outcome after percutaneous mitral valve repair [8,9]. Mehr and coworkers [9] showed in the TRAMI (Transcatheter Mitral Valve Interventions) registry that simultaneous mitral and tricuspid valve transcatheter repair was associated with a higher one-year survival rate compared with isolated transcatheter mitral valve repair in combined mitral regurgitation and TR. However, optimal patient selection for transcatheter edge-to-edge tricuspid valve repair is less known [9,110,111]. Kresoja and coworkers [19] elegantly demonstrated the significance of accurate RV function analysis in patients undergoing transcatheter tricuspid valve repair by using CMR-derived RVEF and 2D echocardiography-derived TAPSE measurement. The results proved that TAPSE was not associated with increased mortality and patients with reduced TAPSE (reduced RV longitudinal function), but RVEF > 45% did not have worse outcomes due to the compensation of circumferential function [19]. The outcome was worse only in global RV dysfunction (RV EF < 45%) when both the longitudinal and circumferential RV functions were diminished [19]. Similarly, Karam N and coworkers [112] showed that conventional RV echocardiographic parameters (TAPSE and sPAP) did not predict clinical outcome after transcatheter tricuspid valve repair. These results emphasize the importance of accurate RV function analysis using RV EF assessment in decision-making and demonstrate that RV longitudinal dysfunction is common and does not always predict adverse outcomes [19,112]. 

Two-dimensional echocardiography-derived vena contracta diameter, PISA-derived EROA, and 3D-echocardiography-derived vena contracta area can be used to quantify the recently proposed severe, massive, and torrential grades of TR, respectively [12,110,111]. Notably, the 2D-echocardiography-derived PISA method is based on the assumption of symmetric and round flow convergence [113]. Previous 3D echocardiography studies showed that the flow convergence of TR was often elliptical or complex [113]. Direct 3D planimetry analysis of vena contracta area can improve the accuracy of even complex TR quantification; resolution and color Doppler blooming are nonetheless the main limitations of accuracy [113] (Figure 4 and Figure 5). Despite the limitations, a large number of publications showed good correlation between the 2D echocardiography-derived PISA method and the 3D echocardiography-derived planimetry method, mainly in patients with moderate to severe TR and in patients with sinus rhythm and without pacemaker leads [94,95,113,114]. However, Abudiab and coworkers [115] found that 3D vena contracta area was superior to 2D PISA for determining TR severity. It is still debatable whether TR severity itself, or rather in combination with RV dysfunction or PH or left ventricular dysfunction, accounts for the outcome in TR population [113]. Furthermore, three-dimensional echocardiography might facilitate a more accurate assessment of the tricuspid valve and surrounding tissue anatomy and pathology, which represents an added value of 3D echocardiography compared to 2D echocardiography. (Figure 6) Previous studies reported contradictory results about RVEF in ASD patients, probably as a consequence of inhomogeneity in the study population in terms of shunt volume, TR presence, and PH degree [40,41,116]. Vitarelli and coworkers [40] showed that 3D RVEF was normal in ASD patients with mildly elevated pulmonary artery pressures, whereas 3D RVEF decreased with severe PH. Transcatheter ASD closure is the first-line therapy in secundum ASD [117]. Optimal patient selection is of utmost importance, as symptoms and RV dilatation may persist after transcatheter ASD closure in severe PH [40]. The 3D RVEF and RV strain assessments proved to be better predictors of adverse outcomes after ASD closure compared to 2D echocardiography parameters [40]. Therefore, accurate assessment of RV function may improve risk assessment and appropriate timing of treatment for ASD [40]. Furthermore, 3D transesophageal echocardiography enables a more accurate definition of ASD size and its relation to surrounding tissue, which facilitates optimal patient selection and operative planning for ASD closure [10,118]. Severe pulmonary regurgitation is a frequent cause of progressive RV dilatation and dysfunction in repaired TOF patients [10,49]. Accurate RV assessment is crucial for indicating the most appropriate timing for pulmonary valve replacement [10,49]. CMR is ideal for accurate quantification of RV volumes and EF; however, 3D echocardiography might help with regular follow-up evaluation and longitudinal RV measurements in this patient population [10]. Furthermore, 3D strain measurements in repaired TOF population revealed that the relative contribution of the longitudinal component to global RVEF was more prominent than either the radial or the anteroposterior shortening [119]. 

### 6.3. Intrinsic RV Myocardial Disease 

Accurate assessment of RV function is important in patients with acute myocardial infarction. Kidawa and coworkers [120] demonstrated that a threshold of 3D RVEF < 51% might be used to diagnose RV myocardial infarction with a sensitivity of 91% and a specificity of 80%. However, 3D RVEF did not perform better than tissue Doppler imaging-derived S’ wave in the diagnosis of RV myocardial infarction [120]. In a study of non-ischemic dilated cardiomyopathy, 3D RVEF was the only independent predictor of adverse outcome after adjusting for age, NYHA class, ratio of early diastolic transmitral flow velocity to tissue Doppler mitral annular early diastolic velocity E/E’, and left atrial volume index [121]. Patients with a value of 3D RVEF < 43.4% were associated with an adjusted threefold-increased risk of major adverse cardiovascular events [121]. Furthermore, the added value of 3D RVEF assessment was highlighted after cardiac surgery, when conventional indexes of RV longitudinal function were usually low due to the postcardiotomy state [122]. The challenging diagnosis of ARVC is based on Task Force Criteria which require assessment of RV regional wall motion abnormalities and RV diameters [123]. In clinical practice, 2D echocardiography and CMR are used to determine if these criteria are met [123,124]. Addetia and coworkers [124] found that the combination of 2D and 3D echocardiography in the diagnosis of ARVC was comparable to the combination of 2D echocardiography and CMR. Cardiac amyloidosis and deposition of amyloid protein may affect both ventricles, usually mainly at the basal segments [125]. Vitarelli and coworkers [125] demonstrated that combining the triad of 3D speckle-tracking echocardiography-derived basal left ventricular longitudinal strain, left ventricular peak basal rotation, and basal RV longitudinal strain might help in differentiating cardiac amyloidosis from other forms of myocardial hypertrophy with a specificity of 86% and a sensitivity of 92%. Consequently, 3D echocardiography might be useful when CMR is not applicable [125]. The reduction of RV 3D basal longitudinal strain was more prominent, although the RV apical sparing pattern was less pronounced compared to the left ventricle in amyloidosis, which might be explained by the complexity of RV geometry and different myofiber orientation [125]. 

Anthracyclines have been reported as potential dose-dependent, irreversible cardiotoxic drugs to treat cancer, leading to chemotherapy-related cardiac dysfunction (CTRCD) [126,127]. Zhao and coworkers [128] found that subclinical changes in RV function and size appeared after the fourth cycle of anthracyclin therapy using 3D echocardiography-based RV volume and function assessment. In the study by Zhao and coworkers [128], the RV cardiotoxicity was determined as a >10% relative reduction in RVEF or a relative reduction of >5% to an absolute value of 45%. After the completion of the fourth cycle of anthracyclin, 3D RVESV, 3D RV-free wall longitudinal strain, and 3D left ventricular global longitudinal strain decreased significantly, whereas 3D RVEF decreased only after the sixth cycle [128]. Notably, baseline 3D RV parameters were not predictors of cardiotoxicity [128].

## 7. Future Directions in Echocardiography to Assess the Right Ventricle

Three-dimensional echocardiography opens future directions in right ventricular assessment, including RV shape analysis, RV segmental analysis, RV strain, and myocardial work analysis. The three-dimensional global RV function is determined by longitudinal, radial, and anteroposterior motion components; however, the relative contributions of these motion components are usually not quantified [129]. Previously, it was thought that longitudinal RV shortening of the subendocardial myocytes is dominant, which accounts for approximately 75% of the right ventricular contraction [20,26,99]. Consequently, the radial and anteroposterior components of RV motion were neglected [99]. However, recently published studies suggest that radial and anteroposterior right ventricular shortenings have comparable significance to longitudinal shortening in determining global RV function [129]. Lakatos and coworkers [129] found that anteroposterior and longitudinal components are the most prominent motion directions of RV function in healthy populations. In addition, the working group revealed an age-dependent shift until the age of 60 years with decreasing longitudinal RV shortening and increasing radial motion [129]. Furthermore, anteroposterior shortening of RV might represent the effect of LV contraction on RV (LV-RV interaction) [129]. Notably, the relative contribution of the three RV motion directions might shift differently in different pathophysiologic states, resulting in the same 3D echocardiography-derived global RV EF value [100,129]. Normal 3D-derived RV EF does not always mean a normal contraction pattern [87]. Postcardiac surgery and heart transplant patients show reduced longitudinal RV function as measured by TAPSE despite normal RV EF [130]. Furthermore, longitudinal RV function can be diminished in volume overload such as pulmonary regurgitation and atrial septal defect [100]. Meanwhile, the radial function might be lower in pressure overloads such as PH and acute pulmonary embolism [100]. Nonetheless, further studies are needed to evaluate the added prognostic and diagnostic value of RV motion direction shift in different health conditions. 

Similar to the left ventricular segmental analysis, a more detailed segmental RV mechanical pattern analysis might enable the detection of subtle segmental dysfunction and improve our diagnostic knowledge of RV pathology [131]. Ishizu and coworkers [132] showed that segmental deformations affect global RV function differently: inlet area strain and outflow circumferential strain was significantly associated with RV EF. Addetia and coworkers [133] established normal three-dimensional echocardiographic values of RV regional curvature index in healthy subjects and separated six right ventricular regions including the inflow tract, the outflow tracts, the septal and the free-wall body, and the septal and free-wall apex [133]. The apical free wall was convex and the septum was more concave, compared to the body-free wall. During RV contraction, the inflow, the outflow tract, and the body-free wall became flatter, while the apex-free wall became more convex [133]. The authors found that the right ventricle is stiffer in older subjects, with less dynamic contraction of the inflow tract and less bellows-like movement; however, no gender differences could be observed [133]. Satriano and coworkers [134] showed that pulmonary hypertension impairs mainly the free RV wall segments. Li and coworkers [135] reported that both 2D and 3D RV longitudinal strain parameters were significant predictors of adverse outcomes in the pulmonary artery hypertension population. Overall, a three-dimensional assessment of the right ventricular shape might represent a future direction in clinical practice, as different pathophysiological states are associated with different maladaptive remodeling [80]. In the current daily routine, the shape of the RV is characterized only by a 2D echocardiography-derived eccentricity index, which enables the separation between right ventricular pressure and volume overload. 

Despite the prognostic value of RV longitudinal strain, it is a more afterload-dependent parameter compared to LV global longitudinal strain [136,137]. Right ventricular myocardial work is a further novel method for non-invasive RV assessment using RV pressure–strain loops evaluated from speckle tracking echocardiography-derived RV global longitudinal strain and noninvasive brachial cuff blood pressure measurements [136]. Butcher and coworkers [136] found that RV global constructive work correlates with the invasively measured right ventricular stroke volume and stroke volume index. Furthermore, right ventricular global constructive work was associated with all-cause mortality in patients with pulmonary hypertension [138]. Right ventricular myocardial work could be also used to assess RV function in the ASD population and might be superior to load-dependent RV GLS [139]. In an experimental rat model, Ebata and coworkers [140] demonstrated that in pulmonary hypertension, the RV lateral wall work is asymmetrically higher, while in pulmonary regurgitation, both lateral and septal work are higher compared to the control group. In addition, the working group showed that asymmetric RV work and increased wasted septal work are associated with RV fibrosis and dysfunction [140]. 

## 8. Conclusions

The prognostic implications of RV function and TR severity emphasize the need for accurate assessment. Three-dimensional echocardiography is essential for the precise evaluation of RV and TV anatomy and function [141]. Despite guidelines, TAPSE is currently the most commonly used RV functional parameter in routine echocardiography, representing only the longitudinal RV function [19,112]. RV longitudinal dysfunction is frequent and does not always predict adverse outcomes [19,112]. The circumferential RV function may compensate for RV longitudinal dysfunction at an earlier stage; consequently, the RVEF remains normal. Three-dimensional echocardiography imaging allows a more accurate quantitative and qualitative assessment of the right ventricle and TR, thereby enabling a more in-depth pathophysiological understanding of the right ventricular function and different TR phenotypes, which may help clinicians in treatment decisions [19,112]. The prognostic value of 3D echocardiography-derived RV functional parameters proved to be superior in a large number of clinical scenarios compared to 2D echocardiography [141]. Quantitative and qualitative 3D assessment of tricuspid regurgitation and valve anatomy improves clinical management, from accurate diagnosis and risk assessment to optimal treatment selection. Advances in cardiac imaging, with the incorporation of machine learning algorithms in 3D echocardiography provide a more reliable, feasible, faster, and user-friendly examination of the right ventricle, which might facilitate its widespread use in daily clinical practice.

## Figures and Tables

**Figure 1 diagnostics-13-02470-f001:**
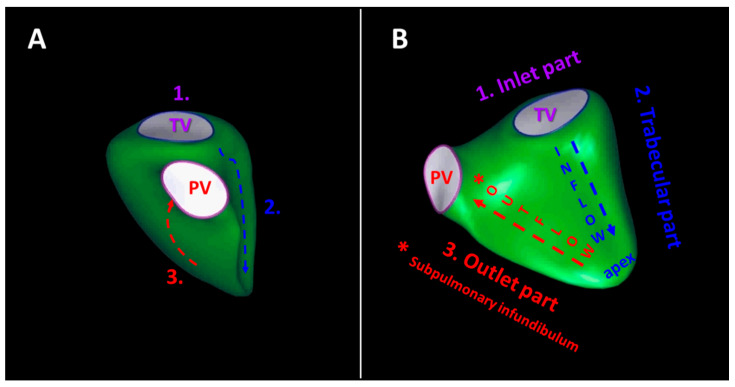
Schematic illustration of the right ventricular components (**1., 2., 3.**) from anterior (panel (**A**)) and septal (panel (**B**)) perspectives. The inlet part (**1.**) constitutes the tricuspid valve apparatus, the trabecular part (**2.**) involves the apex with the three intracavitary muscle bands, and the outlet part (**3**.) includes the subpulmonary infundibulum (*). The contraction of the right ventricle starts earlier within the inlet and trabeculated myocardium than the outlet myocardium.

**Figure 2 diagnostics-13-02470-f002:**
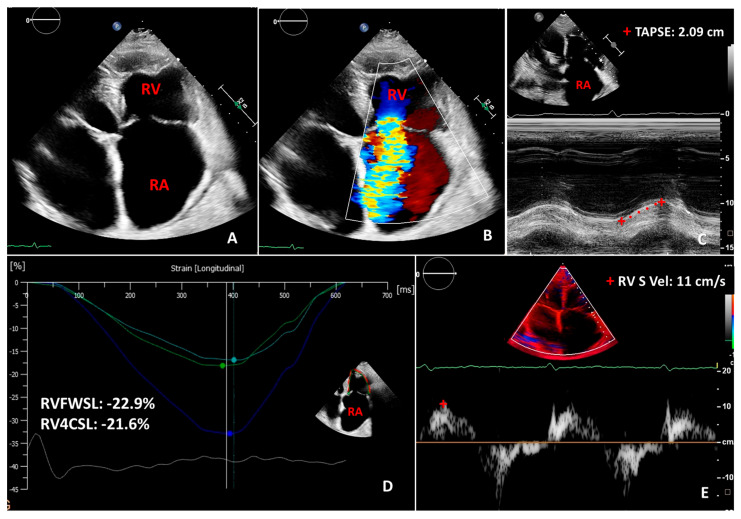
Two-dimensional transthoracic echocardiography images in a case of severe functional tricuspid regurgitation with enlarged atria (**A**,**B**). Panels (**C**–**E**) show normal two-dimensional longitudinal indices of right ventricular function, such as tricuspid annular planar systolic excursion (TAPSE), right ventricular free wall longitudinal strain (RVFWSL), right ventricular 4-chamber longitudinal strain (RV4CSL), right ventricular tissue doppler imaging-derived systolic (S) wave velocity (RV S vel). RA: right atrium; RV: right ventricle.

**Figure 3 diagnostics-13-02470-f003:**
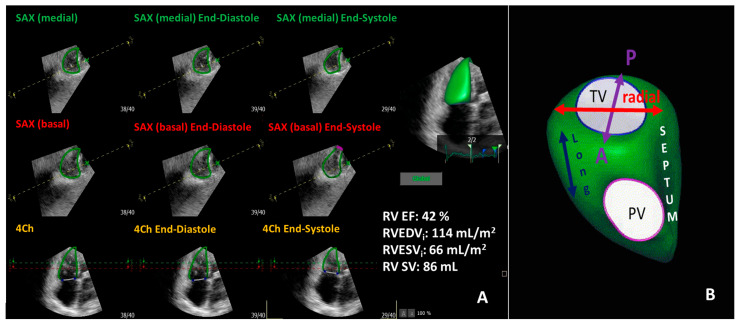
Representative three-dimensional transthoracic echocardiography evaluation of right ventricular function (panel (**A**)) showing mildly diminished right ventricular function in the same patient in Figure 1. Panel (**B**) demonstrates the three-direction motion components of right ventricular function. RV EF: right ventricular ejection fraction; RVEDVi: indexed right ventricular end-diastolic volume; RVESVi: indexed right ventricular end-systolic volume; RV SV: right ventricular stroke volume; TV: tricuspid valve; PV: pulmonic valve; Long: longitudinal; SAX: short axis; 4Ch: four chamber.

**Figure 4 diagnostics-13-02470-f004:**
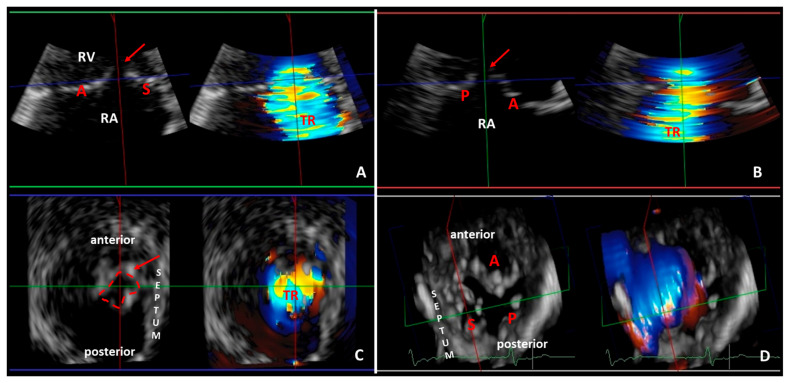
Representative three-dimensional transthoracic echocardiography analysis showing severe functional tricuspid regurgitation with significant coaptation gap (red arrow) in the same patient of Figure 1 and Figure 2. The three-dimensional data set of the tricuspid valve is analyzed with multiplanar reformation planes that display the valve in three orthogonal planes: green (panel **A**), red (panel **B**) and blue (panel **C**) planes). The green (panel **A**) and red (panel **B**) planes cross the tricuspid valve longitudinally. The blue plane (panel **C**) crosses the tricuspid valve at the level of the leaflets coaptation and shows the short-axis of the valve from right ventricular perspective. (Panel **D**) demonstrates the three leaflets of the tricuspid valve in a surgical view (from atrial perspective): septal leaflet (S); anterior leaflet (A); and posterior leaflet (P). RA: right atrium, RV: right ventricle.

**Figure 5 diagnostics-13-02470-f005:**
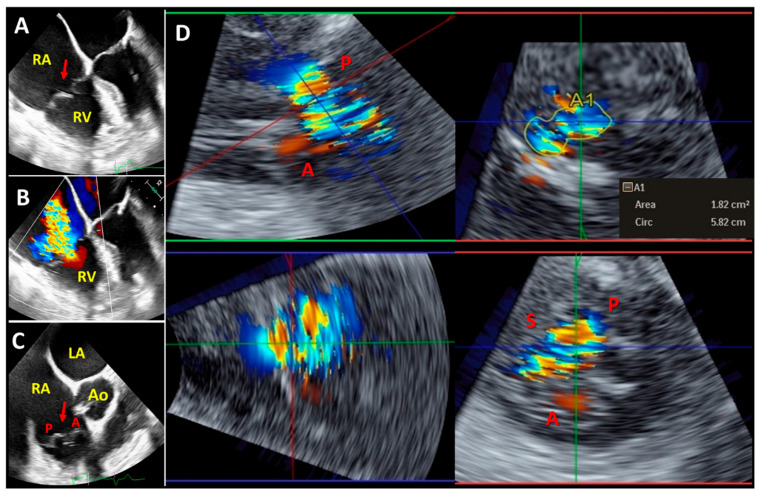
Representative two-dimensional mid-esophageal ((**A**–**C**) panels) and three-dimensional transgastric (**D**) panel) echocardiography images showing severe tricuspid regurgitation and prolapsing tricuspid valve (red arrow). (Panel **D**) demonstrates a three-dimensional multiplanar reconstruction of the tricuspid valve acquired from transgastric right ventricular long-axis view and three-dimensional vena contracta area (A1) measurement. LA: left atrium; RA: right atrium; RV: right ventricle; Ao: aortic valve; S: septal leaflet; A: anterior leaflet; P: posterior leaflet.

**Figure 6 diagnostics-13-02470-f006:**
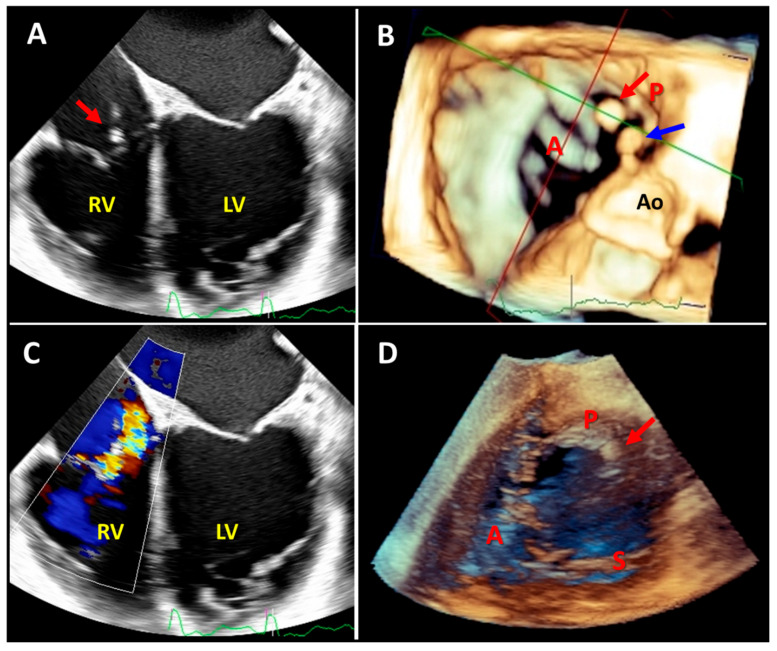
Two-dimensional mid-to-deep esophageal (panels (**A**,**C**)), three-dimensional mid-to-deep esophageal (panel (**B**)), and three-dimensional transgastric echocardiography images demonstrating the added value of three-dimensional echocardiography after pacemaker lead extraction due to infection. Two-dimensional echocardiography shows one mobile mass on the posterior leaflet of the tricuspid valve (panel (**A**), red arrow) and moderate tricuspid regurgitation (panel (**C**)); however, three-dimensional echocardiography revealed a further mass (panel (**B**), blue arrow) on the right atrial wall adjacent to the sinus of Valsalva. Notably, the transgastric three-dimensional view (panel (**D**)) showed only the mass attached to the posterior leaflet (red arrow). RV: right ventricle, LV: left ventricle, Ao: aortic valve, A: anterior leaflet, P: posterior leaflet, S: septal leaflet.

## Data Availability

Not applicable.

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
