# Peer review of "Echocardiography Imaging of the Right Ventricle: Focus on Three-Dimensional Echocardiography"

_diagnostics, 2023, doi:10.3390/diagnostics13152470_

Round 1
Reviewer 1 Report
1. please add the 3D TEE imaging other than 3D RVEF.
2. would please show the image demonstrating accurate mechanism or pathophysiology of TR using 3D TEE?
nothing to comment
Author Response
RESPONSE TO REVIEWER #1
We would like to express our thanks to Reviewer#1 for the careful evaluation of the manuscript and the helpful and constructive suggestions. We have heeded the Reviewer’s helpful propositions and prepared a revised version of the manuscript, which includes the alterations suggested by the Reviewer. Please find our point-by-point responses to the comments below.
Comment/suggestion 1: “please add the 3D TEE imaging other than 3D RVEF.”
Response to Comment 1: Thank You for the suggestion. Accordingly, the figures have been completely changed. Figure 1 illustrates the anatomical parts of the RV. Figure 2. summarizes briefly the most commonly used conventional right ventricular functional 2D echocardiography parameters. Figure 3 is the only figure including 3D RVEF measurements. Figure 4. demonstrates a 3D transthoracic echocardiography evaluation of the tricuspid valve. Figure 5., panel D shows a 3D transesophageal multiplanar reconstruction of the tricuspid valve acquired from transgastric right ventricular long axis view. Figure 6 demonstrates the added value of 3D echocardiography compared to 2D echocardiography, due to its greater appreciation of tricuspid valve and surrounding tissue anatomy and pathology.
Comment/suggestion 2: “would please show the image demonstrating accurate mechanism or pathophysiology of TR using 3D TEE?”
Response to Comment 2: Thank You for the useful comment. Figure 4 demonstrates severe functional tricuspid regurgitation using 3D transthoracic echocardiography. (Notably, Figures 1, 2, 3, and 4 are acquired from the same patient.) Figure 5 shows severe tricuspid regurgitation with prolapsing leaflets using 3D transesophageal echocardiography. Figure 6 presents a case of pacemaker lead extraction due to infection, which emphasizes the added value of 3D echocardiography compared to 2D echocardiography in the evaluation of anatomical and pathological structures.
The new figures are highlighted in yellow in the manuscript and below:
Figure 1. Schematic illustration of the right ventricular components (1., 2., 3.) from anterior (panel A) and septal (panel B) perspectives. The inlet part (1.) constitutes the tricuspid valve apparatus, the trabecular part (2.) involves the apex with the three intracavitary muscle bands, and the outlet part (3.) includes the subpulmonary infundibulum (*). The contraction of the right ventricle starts earlier within the inlet and trabeculated myocardium than the outlet myocardium.
Figure 2. Two-dimensional transthoracic echocardiography in case of severe functional tricuspid regurgitation with enlarged atria (A, B). Two-dimensional longitudinal indices of right ventricular function (C, D, E), such as tricuspid annular planar systolic excursion (TAPSE), right ventricular free wall longitudinal strain (RVFWSL), right ventricular 4- chamber longitudinal strain (RV4CSL), right ventricular tissue doppler imaging-derived systolic (S) wave velocity (RV S vel), show normal right ventricular longitudinal function. RA: right atrium; RV: right ventricle.
Figure 3. Representative three-dimensional transthoracic echocardiography evaluation of right ventricular function (panel A) showing mildly diminished right ventricular function in the same patient in Figure 1. Panel B demonstrates the three-direction motion components of right ventricular function. RV EF: right ventricular ejection fraction; RVEDVi: indexed right ventricular end-diastolic volume; RVESVi: indexed right ventricular end-systolic volume; RV SV: right ventricular stroke volume; TV: tricuspid valve; PV: pulmonic valve; Long: longitudinal.
Figure 4. Representative three-dimensional transthoracic echocardiography images showing severe functional tricuspid regurgitation with significant coaptation gap (red arrow) in the same patient of Figures 1. and 2. Panel D demonstrates the three leaflets of the tricuspid valve in a surgical view: septal leaflet (S); anterior leaflet (A) and posterior leaflet (P). RA: right atrium, RV: right ventricle.
Figure 5. Representative two-dimensional mid-esophageal (A, B, C panels) and three-dimensional transgastric (D panel) echocardiography images showing severe tricuspid regurgitation and pro-lapsing tricuspid valve (red arrow). Panel D demonstrates a three-dimensional multiplanar reconstruction of the tricuspid valve acquired from transgastric right ventricular long-axis view and three-dimensional vena contracta area (A1) measurement. LA: left atrium; RA: right atrium; RV: right ventricle; Ao: aortic valve; S: septal leaflet; A: anterior leaflet; P: posterior leaflet.
Figure 6. Two-dimensional mid-to-deep esophageal (panels A and C), three-dimensional mid-to-deep esophageal (panel B), and three-dimensional transgastric echocardiography images demonstrating the added value of three-dimensional echocardiography after pacemaker lead ex-traction due to infection. Two-dimensional echocardiography shows one mobile mass on the posterior leaflet of the tricuspid valve (panel A, red arrow) and moderate tricuspid regurgitation (panel C), however, three-dimensional echocardiography revealed a further mass (panel B, blue arrow) on the right atrial wall adjacent to Sinus of Valsalva. Notably, the transgastric three-dimensional view (panel D) showed only the mass attached to the posterior leaflet (red arrow). RV: right ventricle, LV: left ventricle, Ao: aortic valve, A: anterior leaflet, P: posterior leaflet, S: septal leaflet.
Once again, we would like to thank the Reviewer for the insightful comments and suggestions! We do believe these resulted in a much-improved manuscript that may be acceptable for publication in Diagnostics.
Please see the attachment.
Andrea Agnes Molnar, MD, PhD

Reviewer 2 Report
General:
Thank you for the opportunity to review this manuscript. The authors provide an overview of the anatomy and pathophysiology of the right ventricle (RV) and the associated tricuspid valve (TV). The manuscript is well written, with the exception of minor typographical and spelling errors, goes into depth on the different mechanisms of RV and TV pathologies, the references are up to date, which overall give the interested reader the opportunity to delve deeper into the subject.
Major comments:
A) Indeed, as the title suggests, the authors tried to focus on three-dimensional (3D) echocardiography in the presentation of RV. However, the manuscript lacks general visual demonstrations between 2D and 3D echocardiography of the RV and TV. Sure, they give examples of normal and reduced RV ejection fraction in Figure 3 and Figure 4, but this is only visible through the parameters provided, calculated by the software. Figure 4 does not give much information. There is no 3D representation of the TV. The authors should present a side-by-side of 2D and 3D comparison in a multi-plot figure for each pathology of RV function and TV mentioned in the text. This would indeed strengthen their extensive literature review and bring their take-home massage closer to the reader who is less familiar with 3D imaging.
B) In addition, authors should give a brief overview of the manuscript in the introduction section, outlying and why it is important to understand the anatomy, function and pathophysiology of the RV in order to understand why 2D imaging is inferior to 3D imaging. Otherwise, the text could be misinterpreted like a recapitulation of a text book before the main part of the manuscript starts: 3D imaging of the RV.
C) In addition, the authors should also give an outlook on the future direction of 3D imaging, such as 3D strain, intraoperative assessment of RV function, and show the resembles of the findings to LV. Are there similar conclusions between 2D and 3D analysis like in the case of LV?
D) Finally, the authors state: "Despite guidelines, TAPSE is currently the most commonly used RV functional parameter in routine echocardiography, representing only longitudinal RV function. " (page 12 line 512). Precisely for this reason, the authors should give some tips and tricks on how to integrate 3D echocardiography into daily practice and which obstacles can be avoided, because clinicians have limited time in routine.
Minor comments:
1) Figure 1: right panel - is it the RV with the strain layer on the right? A side switch would be more helpful for the reader so that the RV is shown on the same side in both the right and left panels.
2) Figure 2: The 2D display of the TV is nice, but how does it relate to the focus on 3D imaging - see comments above on side-by-side comparison of 2D and 3D imaging.
3) Figure 3: adds no information to Figure 2. See comment above.
4) Is it possible to provide an anatomical picture of the RV for the less knowledgeable reader as the authors discuss the function of the myocardial layers and the different anatomical parts of the RV.
5) Chapters 3 and 4 should be more concise, otherwise the authors should consider merging the 2D and 3D chapters and comparing the methods side by side for RV and TV and the corresponding pathologies. This would improve the take home message that 3D is superior to 2D.
Author Response
RESPONSE to REVIEWER #2
We would like to thank the careful review and the helpful suggestions to Reviewer#2. We have heeded the Reviewer’s constructive propositions and prepared a revised version of the manuscript, which includes the alterations suggested by the Reviewer.
Please find our point-by-point responses to the comments below.
Major comments:
Comment A: Indeed, as the title suggests, the authors tried to focus on three-dimensional (3D) echocardiography in the presentation of RV. However, the manuscript lacks general visual demonstrations between 2D and 3D echocardiography of the RV and TV. Sure, they give examples of normal and reduced RV ejection fraction in Figure 3 and Figure 4, but this is only visible through the parameters provided, calculated by the software. Figure 4 does not give much information. There is no 3D representation of the TV. The authors should present a side-by-side of 2D and 3D comparison in a multi-plot figure for each pathology of RV function and TV mentioned in the text. This would indeed strengthen their extensive literature review and bring their take-home massage closer to the reader who is less familiar with 3D imaging.
Response to Comment A: Thank You for the comment! Indeed, the manuscript lacked general visual demonstrations between 2D and 3D echocardiography, therefore we have completely changed the figures according to the suggestions. Figure 2 summarizes the most commonly used conventional 2D RV functional parameters and Figure 3 shows the 3D RV functional parameters of the same patient described in Figure Legend. Furthermore, Figure 4 illustrates a functional and Figure 5 a degenerative tricuspid regurgitation case including 2D and 3D images (the 2D images of the functional tricuspid regurgitation case are placed in Figure 2). In addition, the case of Figure 6 compares side-by-side the 2D and 3D images of a tricuspid valve after pacemaker lead extraction, and demonstrates the added value of 3D echocardiography compared to 2D echocardiography, due to its greater appreciation of tricuspid valve and surrounding tissue anatomy and pathology.
Comment B: In addition, authors should give a brief overview of the manuscript in the introduction section, outlying and why it is important to understand the anatomy, function and pathophysiology of the RV in order to understand why 2D imaging is inferior to 3D imaging. Otherwise, the text could be misinterpreted like a recapitulation of a text book before the main part of the manuscript starts: 3D imaging of the RV.
Response to Comment B: Thank You for the suggestion! We have supplemented the Introduction with a short outline of the manuscript emphasizing and explaining the importance of understanding the anatomy, function, and pathophysiology of the RV in order to recognize why 2D imaging is inferior to 3D imaging. The supplementary sentences are highlighted in yellow in the manuscript and below:
“Understanding the anatomy and pathophysiology of the right heart by using competent imaging tools may help in diagnosing and managing the disease [18,19]. Knowledge of the complex anatomy of the right ventricle allows us to interpret the conventionally used 2D echocardiography parameters more properly and to be convinced of the usefulness of 3D echocardiography in estimating RV size and function. Most of the 2D echocardiography RV functional parameters used in daily clinical practice are one-dimensional parameters, which cannot estimate accurately global RV function. Furthermore, knowledge of RV pathophysiological states with the application of appropriate imaging tools could result in a more accurate diagnosis and optimal patient management. Recently, onsite 3D echocardiography RV analysis became more available in echocardiography laboratories. It is considered less time-consuming, even in routine clinical practice, due to the novel software using artificial intelligence, which has revolutionized data processing and interpretation. However, high-quality image acquisition is still a cornerstone of 3D RV analysis and poor 2D echocardiography image quality cannot be replaced by 3D echocardiography examination.
This review aimed to provide a short overview of the right ventricular and tricuspid valve anatomy, pathophysiology, and 2D/3D echocardiography assessment to demonstrate in detail the added value of 3D echocardiography using the latest published data besides the state-of-the-art literature.”
Comment C: In addition, the authors should also give an outlook on the future direction of 3D imaging, such as 3D strain, intraoperative assessment of RV function, and show the resembles of the findings to LV. Are there similar conclusions between 2D and 3D analysis like in the case of LV?
Response to Comment C: Thank You for the great point! Accordingly, we added a separate Chapter 7 to the manuscript highlighted in yellow:
“7. Future directions in echocardiography to assess the right ventricle
Three-dimensional echocardiography opens future directions in right ventricular assessment including RV shape analysis, RV segmental analysis, RV strain, and myocardial work analysis. The three-dimensional global RV function is determined by longitudinal, radial, and anteroposterior motion components, however, the relative contributions of these motion components are usually not quantified [129]. Previously it was thought, that longitudinal RV shortening of the subendocardial myocytes is dominant, which accounts for approximately 75% of the right ventricular contraction [20,26,99]. Consequently, the radial and anteroposterior components of RV motion were neglected [99]. However, recently published studies suggested that radial and anteroposterior right ventricular shortenings have comparable significance to longitudinal shortening in determining global RV function [129]. Lakatos B and coworkers [129] found that anteroposterior and longitudinal components are the most prominent motion directions of RV function in healthy populations. In addition, the working group revealed an age-dependent shift until the age of 60 years with decreasing longitudinal RV shortening and increasing radial motion [129]. Furthermore, anteroposterior shortening of RV might represent the effect of LV contraction on RV (LV-RV interaction) [129]. Notably, the relative contribution of the three RV motion directions might shift differently in different pathophysiologic states resulting in the same 3D echocardiography-derived global RV EF value [100,129]. Normal 3D-derived RV EF does not always mean a normal contraction pattern [87]. Postcardiac surgery and heart transplant patients show reduced longitudinal RV function as measured by TAPSE despite normal RV EF [130]. Furthermore, longitudinal RV function can be diminished in volume overload such as pulmonary regurgitation and atrial septal defect [100]. Meanwhile, the radial function might be lower in pressure overloads such as PH and acute pulmonary embolism [100]. Nonetheless, further studies are needed to evaluate the added prognostic and diagnostic value of RV motion direction shift in different health conditions.
Similar to the left ventricular segmental analysis, a more detailed segmental RV mechanical pattern analysis might enable the detection of subtle segmental dysfunction and improve our diagnostic knowledge of RV pathology [131]. Ishizu and coworkers [132] showed that segmental deformations affect global RV function differently: inlet area strain and outflow circumferential strain was significantly associated with RV EF. Addetia K and coworkers [133] establish normal three-dimensional echocardiographic values of RV regional curvature index in healthy subjects and separated six right ventricular regions including the inflow tract, the outflow tracts, the septal and the free-wall body, and the septal and free-wall apex [133]. The apical free wall was convex and the septum was more concave, compared to the bod-free wall. During RV contraction, the inflow, the outflow tract, and the body-free wall became flatter, while the apex-free wall became more convex [133]. The authors found that the right ventricle is stiffer in older subjects, with less dynamic contraction of the inflow tract and less bellows-like movement, however, no gender differences could be observed [133]. Satriano and coworkers [134] showed that pulmonary hypertension impairs mainly the free RV wall segments. Li and coworkers [135] reported, that both 2D and 3D RV longitudinal strain parameters were significant predictors of adverse outcomes in the pulmonary artery hypertension population. Overall, a three-dimensional assessment of the right ventricular shape might represent a future direction in clinical practice as different pathophysiological states are associated with different maladaptive remodeling [80]. In the current daily routine, the shape of the RV is characterized only by a 2D echocardiography-derived eccentricity index, which enables the separation between right ventricular pressure and volume overload.
Despite the prognostic value of RV longitudinal strain, it is a more after-load-dependent parameter compared to LV global longitudinal strain [136,137]. Right ventricular myocardial work is a further novel method for non-invasive RV assessment using RV pressure–strain loops evaluated from speckle tracking echocardiography-derived RV global longitudinal strain and noninvasive brachial cuff blood pressure measurements [136]. Butcher and coworkers [136] found that RV global constructive work correlates with the invasively measured right ventricular stroke volume and stroke volume index. Furthermore, right ventricular global constructive work was associated with all-cause mortality in patients with pulmonary hypertension [138]. Right ventricular myocardial work could be also used to assess RV function in the ASD population and might be superior to load-dependent RV GLS [139]. In an experimental rat model, Ebata and coworkers [140] demonstrated that in pulmonary hypertension the RV lateral wall work is asymmetrically higher, while in pulmonary regurgitation both lateral and septal work is higher compared to the control group. In addition, the working group showed, that asymmetric RV work and increased wasted septal work are associated with RV fibrosis and dysfunction [140].”
Right ventricular volume measurement and the calculation of RVEF by 2D echocardiography are not recommended for clinical use due to the inaccuracy of the method. The 2D method uses apical views to the RVEF evaluation, which excludes the RV outflow tract and therefore underestimates RVEF. (Jonas N et al., 2019) Consequently, there is no comparison between 2D RVEF and 3D RVEF, however, previous studies compared echocardiography-derived 3D RVEF with CMR-derived RVEF. (Shimada YJ et al, 2010) Kresoja and coworkers elegantly demonstrated that TAPSE was not associated with increased mortality and patients with reduced TAPSE, but 3D RVEF >45% did not have worse outcomes due to the compensation of circumferential function. (Kresoja KP et al., 2021) The outcome was worse only in global RV dysfunction (RV EF < 45%) when both the longitudinal and circumferential RV functions were diminished. (Kresoja KP et al., 2021) These results suggest that RV longitudinal dysfunction is common and does not always predict adverse outcomes. Furthermore, Nagata and coworkers found that 3D RVEF, rather than 3D left ventricular EF, stratified patients from low to high risk for subsequent cardiac events (Nagata Y et al., 2017)
References:
Jones, N.; Burns, A.T.; Prior, D.L. Echocardiographic Assessment of the Right Ventricle-State of the Art. Heart Lung Circ 2019, 28, 1339-1350, doi:10.1016/j.hlc.2019.04.016.
Shimada, Y.J.; Shiota, M.; Siegel, R.J.; Shiota, T. Accuracy of right ventricular volumes and function determined by three-dimensional echocardiography in comparison with magnetic resonance imaging: a meta-analysis study. J Am Soc Echocardiogr 2010, 23, 943-953, doi:10.1016/j.echo.2010.06.029.
Kresoja, K.P.; Rommel, K.P.; Lücke, C.; Unterhuber, M.; Besler, C.; von Roeder, M.; Schöber, A.R.; Noack, T.; Gutberlet, M.; Thiele, H.; et al. Right Ventricular Contraction Patterns in Patients Undergoing Transcatheter Tricuspid Valve Repair for Severe Tricuspid Regurgitation. JACC Cardiovasc Interv 2021, 14, 1551-1561, doi:10.1016/j.jcin.2021.05.005.
Nagata, Y.; Wu, V.C.; Kado, Y.; Otani, K.; Lin, F.C.; Otsuji, Y.; Negishi, K.; Takeuchi, M. Prognostic Value of Right Ventricular Ejection Fraction Assessed by Transthoracic 3D Echocardiography. Circ Cardiovasc Imaging 2017, 10, doi:10.1161/circimaging.116.005384.
Comment D: Finally, the authors state: "Despite guidelines, TAPSE is currently the most commonly used RV functional parameter in routine echocardiography, representing only longitudinal RV function. " (page 12 line 512). Precisely for this reason, the authors should give some tips and tricks on how to integrate 3D echocardiography into daily practice and which obstacles can be avoided, because clinicians have limited time in routine.
Response to Comment D: Thank You for the suggestion. Indeed, the integration of 3D echocardiography in daily clinical practice is of utmost importance due to its supplementary and added value compared to 2D echocardiography. According to the Reviewer’s suggestion, we supplemented the Introduction part of the manuscript with the following highlighted sentences (rows 39-44):
“Recently, onsite 3D echocardiography RV analysis became more available in echocardiography laboratories. It is considered less time-consuming, even in routine clinical practice, due to the novel software using artificial intelligence, which has revolutionized data processing and interpretation. However, high-quality image acquisition is still a cornerstone of 3D RV analysis and poor 2D echocardiography image quality cannot be replaced by 3D echocardiography examination.”
Minor comments:
Comment 1: Figure 1: right panel - is it the RV with the strain layer on the right? A side switch would be more helpful for the reader so that the RV is shown on the same side in both the right and left panels.
Response to Comment 1: Thank You for the comment. We have restructured all the figures according to the suggestion of the Reviewers. Images of Figure 1 have been deleted and new images are included.
Comment 2: Figure 2: The 2D display of the TV is nice, but how does it relate to the focus on 3D imaging - see comments above on side-by-side comparison of 2D and 3D imaging.
Response to Comment 2: Thank You for the comment. We have restructured all the figures according to the suggestion of the Reviewers. Images of Figure 2 have been deleted and new images are included. We added a further figure to the manuscript (Figure 5), which provides a side-by-side comparison of 2D and 3D imaging emphasizing the added value of 3D echocardiography compared to 2D echocardiography.
Comment 3: Figure 3: adds no information to Figure 2. See comment above.
Response to Comment 3: Thank You for the comment. We have restructured all the figures according to the suggestion of the Reviewers. Images of Figure 3 have been deleted and new images are included.
Comment 4: Is it possible to provide an anatomical picture of the RV for the less knowledgeable reader as the authors discuss the function of the myocardial layers and the different anatomical parts of the RV.
Response to Comment 4: Thank you for your important point! We provided a schematic anatomical picture of the right ventricular components and the order of their contraction in Figure 1. (highlighted in yellow in the manuscript and below):
“Figure 1. Schematic illustration of the right ventricular components (1., 2., 3.) from anterior (panel A) and septal (panel B) perspectives. The inlet part (1.) constitutes the tricuspid valve apparatus, the trabecular part (2.) involves the apex with the three intracavitary muscle bands, and the outlet part (3.) includes the subpulmonary infundibulum (*). The contraction of the right ventricle starts earlier within the inlet and trabeculated myocardium than the outlet myocardium.”
Comment 5.: Chapters 3 and 4 should be more concise, otherwise the authors should consider merging the 2D and 3D chapters and comparing the methods side by side for RV and TV and the corresponding pathologies. This would improve the take home message that 3D is superior to 2D.
Response to Comment 5.: Thank You for the comment! Accordingly, we have deleted the following sentences from Chapter 4:
“Notably, RV volume measurement and the calculation of RVEF by 2D echocardiography are not recommended for clinical use due to the inaccuracy of the method. The 2D method uses apical views to the RVEF evaluation, which excludes the RV outflow tract and therefore underestimates RVEF. (26)”
“It is the ratio between the sum of isovolumic contraction and relaxation times (representing the tricuspid valve closure to open time or non-ejection time), and the ejection time measured by tissue Doppler method using the standard apical four-chamber view [61]. Furthermore, the pulsed Doppler method uses tricuspid valve inflow waves obtained from apical four-chamber view to measure tricuspid valve closure to open time, and parasternal short axis view to measure the ejection time of blood through the pulmonic valve.”
Once again, we would like to thank you for your insightful comments and suggestions! We do believe these resulted in a much-improved manuscript that may be acceptable for publication in Diagnostics.
Please see the attachment.
Andrea Agnes Molnar, MD, PhD

Reviewer 3 Report
Molnar et al. Written a relatively cursory review concerning right ventricle echo.
The biggest issue of this paper is that it offers no novelty on the topic whatsoever. This could be useful text for someone who wants to enter the “realm” of 3D echo, but it is simply not suitable to be considered for publication as a review article as there are many (more informative) state of the art reviews on this topic.
The authors should shorten the parts concerning classical echo, as the reader should be well aware of that part.
The review would largely benefit from some novel perspective/future directions.
Minor point. TAPSE is not well represented in the figure 1.
Acceptable.
Author Response
RESPONSE to REVIEWER #3
We would like to thank the careful evaluation of the manuscript to Reviewer#3. We have heeded the Reviewer’s comments and prepared a revised version of the manuscript, which includes the alterations suggested by the Reviewer.
Please find our point-by-point responses to the comments below.
Comment 1: The biggest issue of this paper is that it offers no novelty on the topic whatsoever. This could be useful text for someone who wants to enter the “realm” of 3D echo, but it is simply not suitable to be considered for publication as a review article as there are many (more informative) state of the art reviews on this topic.
Response to Comment 1: We understand that a number of issues were raised and therefore, the manuscript was not found suitable for publication in its present form as a review article. We are confident that the raised issues can be resolved through careful revision. Thus, we have prepared a revised version of the manuscript, which includes new figures and a chapter, as well as alterations, as suggested by the Reviewer. This review aimed to provide a short overview of the right ventricular and tricuspid valve anatomy, pathophysiology, and 2D/3D echocardiography assessment to demonstrate in detail the added value of 3D echocardiography using the latest published data besides the state-of-the-art literature. Understanding the anatomy and pathophysiology of the right heart by using competent imaging tools may help in diagnosing and managing the disease. Knowledge of the complex anatomy of the right ventricle allows us to interpret the conventionally used 2D echocardiography parameters more properly and to be convinced of the usefulness of 3D echocardiography in estimating RV size and function.
Comment 2: The authors should shorten the parts concerning classical echo, as the reader should be well aware of that part.
Response to Comment 2: Thank You for the comment! We have summarized briefly the 2D echocardiography of the right ventricle and tricuspid valve to better demonstrate and explain the added value of 3D echocardiography. According to the Reviewer’s suggestion, we have deleted the following sentences from Chapter 4:
“Notably, RV volume measurement and the calculation of RVEF by 2D echocardiography are not recommended for clinical use due to the inaccuracy of the method. The 2D method uses apical views to the RVEF evaluation, which excludes the RV outflow tract and therefore underestimates RVEF. (26)”
“It is the ratio between the sum of isovolumic contraction and relaxation times (representing the tricuspid valve closure to open time or non-ejection time), and the ejection time measured by tissue Doppler method using the standard apical four-chamber view [61]. Furthermore, the pulsed Doppler method uses tricuspid valve inflow waves obtained from apical four-chamber view to measure tricuspid valve closure to open time, and parasternal short axis view to measure the ejection time of blood through the pulmonic valve.”
Comment 3: The review would largely benefit from some novel perspective/future directions.
Response to Comment 3: Thank You for the constructive suggestion! Accordingly, we have supplemented the manuscript with a new chapter discussing the future directions. The new chapter is highlighted in yellow:
“7. Future directions in echocardiography to assess the right ventricle
Three-dimensional echocardiography opens future directions in right ventricular assessment including RV shape analysis, RV segmental analysis, RV strain, and myocardial work analysis. The three-dimensional global RV function is determined by longitudinal, radial, and anteroposterior motion components, however, the relative contributions of these motion components are usually not quantified [129]. Previously it was thought, that longitudinal RV shortening of the subendocardial myocytes is dominant, which accounts for approximately 75% of the right ventricular contraction [20,26,99]. Consequently, the radial and anteroposterior components of RV motion were neglected [99]. However, recently published studies suggested that radial and anteroposterior right ventricular shortenings have comparable significance to longitudinal shortening in determining global RV function [129]. Lakatos B and coworkers [129] found that anteroposterior and longitudinal components are the most prominent motion directions of RV function in healthy populations. In addition, the working group revealed an age-dependent shift until the age of 60 years with decreasing longitudinal RV shortening and increasing radial motion [129]. Furthermore, anteroposterior shortening of RV might represent the effect of LV contraction on RV (LV-RV interaction) [129]. Notably, the relative contribution of the three RV motion directions might shift differently in different pathophysiologic states resulting in the same 3D echocardiography-derived global RV EF value [100,129]. Normal 3D-derived RV EF does not always mean a normal contraction pattern [87]. Postcardiac surgery and heart transplant patients show reduced longitudinal RV function as measured by TAPSE despite normal RV EF [130]. Furthermore, longitudinal RV function can be diminished in volume overload such as pulmonary regurgitation and atrial septal defect [100]. Meanwhile, the radial function might be lower in pressure overloads such as PH and acute pulmonary embolism [100]. Nonetheless, further studies are needed to evaluate the added prognostic and diagnostic value of RV motion direction shift in different health conditions.
Similar to the left ventricular segmental analysis, a more detailed segmental RV mechanical pattern analysis might enable the detection of subtle segmental dysfunction and improve our diagnostic knowledge of RV pathology [131]. Ishizu and coworkers [132] showed that segmental deformations affect global RV function differently: inlet area strain and outflow circumferential strain was significantly associated with RV EF. Addetia K and coworkers [133] establish normal three-dimensional echocardiographic values of RV regional curvature index in healthy subjects and separated six right ventricular regions including the inflow tract, the outflow tracts, the septal and the free-wall body, and the septal and free-wall apex [133]. The apical free wall was convex and the septum was more concave, compared to the bod-free wall. During RV contraction, the inflow, the outflow tract, and the body-free wall became flatter, while the apex-free wall became more convex [133]. The authors found that the right ventricle is stiffer in older subjects, with less dynamic contraction of the inflow tract and less bellows-like movement, however, no gender differences could be observed [133]. Satriano and coworkers [134] showed that pulmonary hypertension impairs mainly the free RV wall segments. Li and coworkers [135] reported, that both 2D and 3D RV longitudinal strain parameters were significant predictors of adverse outcomes in the pulmonary artery hypertension population. Overall, a three-dimensional assessment of the right ventricular shape might represent a future direction in clinical practice as different pathophysiological states are associated with different maladaptive remodeling [80]. In the current daily routine, the shape of the RV is characterized only by a 2D echocardiography-derived eccentricity index, which enables the separation between right ventricular pressure and volume overload.
Despite the prognostic value of RV longitudinal strain, it is a more after-load-dependent parameter compared to LV global longitudinal strain [136,137]. Right ventricular myocardial work is a further novel method for non-invasive RV assessment using RV pressure–strain loops evaluated from speckle tracking echocardiography-derived RV global longitudinal strain and noninvasive brachial cuff blood pressure measurements [136]. Butcher and coworkers [136] found that RV global constructive work correlates with the invasively measured right ventricular stroke volume and stroke volume index. Furthermore, right ventricular global constructive work was associated with all-cause mortality in patients with pulmonary hypertension [138]. Right ventricular myocardial work could be also used to assess RV function in the ASD population and might be superior to load-dependent RV GLS [139]. In an experimental rat model, Ebata and coworkers [140] demonstrated that in pulmonary hypertension the RV lateral wall work is asymmetrically higher, while in pulmonary regurgitation both lateral and septal work is higher compared to the control group. In addition, the working group showed, that asymmetric RV work and increased wasted septal work are associated with RV fibrosis and dysfunction [140].”
Comment 4: Minor point. TAPSE is not well represented in the figure 1.
Response to Comment 4: Thank You for the point! We have changed the figure. Please find below the new figure and the figure legend highlighted in yellow:
“Figure 2. Two-dimensional transthoracic echocardiography images in case of severe functional tricuspid regurgitation with enlarged atria (A, B). Two-dimensional longitudinal indices of right ventricular function (C, D, E), such as tricuspid annular planar systolic excursion (TAPSE), right ventricular free wall longitudinal strain (RVFWSL), right ventricular 4- chamber longitudinal strain (RV4CSL), right ventricular tissue doppler imaging-derived systolic (S) wave velocity (RV S vel), show normal right ventricular longitudinal function. RA: right atrium; RV: right ventricle.”
Once again, we would like to thank the Reviewer for the in-depth review and constructive comments! We do believe these resulted in a much-improved manuscript that may be acceptable for publication in Diagnostics.
Please see the attachment.
Andrea Agnes Molnar, MD, PhD

Reviewer 4 Report
General comment
Understanding the anatomy and pathophysiology of the right heart may help in diagnosing and managing the disease by using reliable imaging tools. Advances in cardiac imaging and three-dimensional echocardiography provided more reliable information on right ventricular volumes and function without geometrical assumptions. Enface 3D views of the tricuspid valve can supplement 2D echocardiography measurements and provide further valuable data regarding leaflets, annular size, etiology and severity of valve regurgitation. The aim of this review was to provide an overview on the clinically relevant pathophysiology and echocardiography assessment of the right ventricle and tricuspid valve with the focus on the significance of 3D echocardiography.
Specific points
1-I think, the authors should write reference numbers after "et al." in the manuscript.
-Hahn RT and coworkers found that the most common anatomic variant is the classic three-leaflet morphology tricuspid valve, which occurs in 28% to 58% [17].
2- I think, the authors should write “RV adaptation” instead of “RV adaption”.
This results in a greater RV adaption to volume overload rather than pressure overload [20].
3- I think, the authors should not use abbreviation in in the titles (e.g.)
-3.1. Acute RV pressure and volume overload
4-The authors misspelled the punctuation mark in the sentences below.
-Pulmonary arterial pressure (PAP) and pulmonary vascular resistance (PVR) increase slowly in chronic pressure overload, consequently the stimulated myocytes lead to adaptive hypertrophy to preserve cardiac output. [31]
-Chronic RV volume overload develops mainly in TR, pulmonary regurgitation and left-to right congenital shunts such as atrial septal defects (ASD). [36,48,49]
-According to the tricuspid leaflet involvement, TR is classified into primary (organic), secondary (functional) and cardiac implantable electronic device (CIED) related regurgitation. [17,50]
-The tricuspid annulus dilates in right atrial or right ventricular enlargement,.
5- I think, the authors should write “systemic diseases” instead of “systematic diseases”.
-Acquired causes comprise endocarditis, prolapse, connective tissue disorder, systematic diseases, radiation, rheumatic disease, tumors and drug-induced leaflet damage. [17,52]
6-The authors should not use abbreviations at the beginning of sentences and lines (e.g.)
-TAPSE has prognostic value in PH and heart failure [66,67].
-RV end-systolic area (ESA) and end-diastolic area (EDA) are used to calculate fractional area change (FAC) and are obtained by tracing the ventricular endocardium in the RV-focused apical four-chamber view.
-RV index of myocardial performance (RIMP) is a load-dependent parameter, which estimates both RV systolic and diastolic function.
-2D speckle tracking echocardiography parameters of RV are less load- and angle-dependent and estimate RV myocardial function more accurately, although temporal resolution is lower.
7-I think, the authors should write “average” instead of “avarage”.
RV-focused four-chamber view is used to measure RV-free wall strain and RV longitudinal strain (avarage of the three free wall and three septal segments).
8-The references should be written according to the writing rules of the journal.
Minor editing of English language required
Author Response
RESPONSE TO REVIEWER #4
We would like to thank the careful evaluation of the manuscript to Reviewer#4. We have heeded the Reviewer’s propositions and prepared a revised version of the manuscript, which includes the alterations suggested by the Reviewer.
Please find our point-by-point responses to the specific points below.
Specific points
Point 1-I think, the authors should write reference numbers after "et al." in the manuscript.
-Hahn RT and coworkers found that the most common anatomic variant is the classic three-leaflet morphology tricuspid valve, which occurs in 28% to 58% [17].
Response to Point 1: Thank You for the suggestion! We have written the reference number after “et al.” in the manuscript, as suggested by the Reviewer.
Point 2- I think, the authors should write “RV adaptation” instead of “RV adaption”.
This results in a greater RV adaption to volume overload rather than pressure overload [20].
Response to Point 2: Thank You for the correction! We have changed the words.
Point 3- I think, the authors should not use abbreviation in in the titles (e.g.)
-3.1. Acute RV pressure and volume overload
Response to Point 3: Thank you for the comment! We have corrected the abbreviations.
Point 4-The authors misspelled the punctuation mark in the sentences below.
-Pulmonary arterial pressure (PAP) and pulmonary vascular resistance (PVR) increase slowly in chronic pressure overload, consequently the stimulated myocytes lead to adaptive hypertrophy to preserve cardiac output. [31]
-Chronic RV volume overload develops mainly in TR, pulmonary regurgitation and left-to right congenital shunts such as atrial septal defects (ASD). [36,48,49]
-According to the tricuspid leaflet involvement, TR is classified into primary (organic), secondary (functional) and cardiac implantable electronic device (CIED) related regurgitation. [17,50]
-The tricuspid annulus dilates in right atrial or right ventricular enlargement,.
Response to Point 4: Thank you for the comment! We have corrected the punctuation marks.
Point 5- I think, the authors should write “systemic diseases” instead of “systematic diseases”.
-Acquired causes comprise endocarditis, prolapse, connective tissue disorder, systematic diseases, radiation, rheumatic disease, tumors and drug-induced leaflet damage. [17,52]
Response to Point 5: Thank You for the point! We have corrected the word.
Point 6-The authors should not use abbreviations at the beginning of sentences and lines (e.g.)
-TAPSE has prognostic value in PH and heart failure [66,67].
-RV end-systolic area (ESA) and end-diastolic area (EDA) are used to calculate fractional area change (FAC) and are obtained by tracing the ventricular endocardium in the RV-focused apical four-chamber view.
-RV index of myocardial performance (RIMP) is a load-dependent parameter, which estimates both RV systolic and diastolic function.
-2D speckle tracking echocardiography parameters of RV are less load- and angle-dependent and estimate RV myocardial function more accurately, although temporal resolution is lower.
Response to Point 6: Thank You for the suggestion! Accordingly, we have corrected the abbreviations at the beginning of sentences and lines.
Point 7-I think, the authors should write “average” instead of “avarage”.
RV-focused four-chamber view is used to measure RV-free wall strain and RV longitudinal strain (avarage of the three free wall and three septal segments).
Response to Point 7: Thank You for the comment! We have corrected the word.
Point 8-The references should be written according to the writing rules of the journal.
Response to Point 8: Thank You for the point! We have used an EndNote template to manage the references, which was downloaded from the website of DPI.https://www.mdpi.com/authors/references.
Once again, we would like to thank the Reviewer for the specific points and suggestions! We do believe these resulted in a much-improved manuscript that may be acceptable for publication in Diagnostics.
Please see the attachment.
Andrea Agnes Molnar, MD, PhD

Round 2
Reviewer 2 Report
Thank you for the opportunity to review this article.
The authors responded most of my comments.
Very insightful article, especially the last chapter on the future outlook and perspectives of the regional RV function.
Just a minor comment on Figure 6:
Maybe you should emphasize the "aha" moment of the 3D, which does not really exist in 2D. Because it even looks like there is not only a second vegetation on the leaflet, but also a destruction and therefore certainly a torrential/severe TR. Maybe you should emphasize that this can only be shown in 3D. Maybe you should point this out.
Reviewer 3 Report
I thank the authors for their extensive response, they improved the quality of their manuscript. Unfortunately, I still find this review not worthy of publishing.
English language quality is acceptable.